# An Approach for Detecting Mangrove Areas and Mapping Species Using Multispectral Drone Imagery and Deep Learning

**DOI:** 10.3390/s25082540

**Published:** 2025-04-17

**Authors:** Xingyu Chen, Xiuyu Zhang, Changwei Zhuang, Xuejiao Dai, Lingling Kong, Zixia Xie, Xibang Hu

**Affiliations:** 1Institute of Ecological Civilization and Green Development, Guangdong Provincial Academy of Environmental Science, Guangzhou 510045, China; chenxy2019@mail.sustech.edu.cn (X.C.); ecozcw@163.com (C.Z.); xiezx.18s@igsnrr.ac.cn (Z.X.);; 2Ecological Environment Remote Sensing Research Center, Guangdong Provincial Academy of Environmental Science, Guangzhou 510045, China

**Keywords:** mangrove ecosystems, high-resolution multispectral data, mangrove species identification, convolutional neural network, spectral and spatial attention mechanisms

## Abstract

Mangrove ecosystems are important in tropical and subtropical coastal zones, contributing to marine biodiversity and maintaining marine ecological balance. It is crucial to develop more efficient, intelligent, and accurate monitoring methods for mangroves to understand better and protect mangrove ecosystems. This study promotes a novel model, MangroveNet, for integrating multi-scale spectral and spatial information and detecting mangrove area. In addition, we also present an improved model, AttCloudNet+, to identify the distribution of mangrove species based on high-resolution multispectral drone images. These models incorporate spectral and spatial attention mechanisms and have been shown to effectively address the limitations of traditional methods, which have been prone to inaccuracy and low efficiency in mangrove species identification. In this study, we compare the results from MangroveNet with SegNet, UNet, and DeepUNet, etc. The findings demonstrate that the MangroveNet exhibits superior generalization learning capabilities and more accurate extraction outcomes than other deep learning models. The accuracy, F1_Score, mIoU, and precision of MangroveNet were 99.13%, 98.84%, 98.11%, and 99.14%, respectively. In terms of identifying mangrove species, the prediction results from AttCloudNet+ were compared with those obtained from traditional supervised and unsupervised classifications and various machine learning and deep learning methods. These include K-means clustering, ISODATA cluster analysis, Random Forest (RF), Support Vector Machines (SVM), and others. The comparison demonstrates that the mangrove species identification results obtained using AttCloudNet+ exhibit the most optimal performance in terms of the Kappa coefficient and the overall accuracy (OA) index, reaching 0.81 and 0.87, respectively. The two comparison results confirm the effectiveness of the two models developed in this study for identifying mangroves and their species. Overall, we provide an efficient solution based on deep learning with a dual attention mechanism in the acceptable real-time monitoring of mangroves and their species using high-resolution multispectral drone imagery.

## 1. Introduction

Mangroves are one of the most sensitive parts of the marine ecosystem to ecological environment change [1]. The variation in mangroves and their species can assist in understanding the health status of mangrove ecosystems, biodiversity, ecosystem function, and their response to climate change [2,3]. Mangrove forests are widely distributed in tropical coastal zones around the world [4]. Obtaining the range and species distribution of mangrove forests with high spatial and temporal resolution is positively significant for analyzing regional and even global marine ecosystem changes [5,6].

The initial ecological monitoring of mangrove ecosystems predominantly employed artificial field sampling surveys to ascertain approximate mangrove areas and species [7]. However, this approach proved inadequate for determining the spatial distribution of mangrove species [8]. In addition, this method requires substantial financial investment as well as considerable human and temporal resources. Moreover, the monitoring efficiency remains challenging. The advent of satellite remote sensed data saw researchers begin to analyze and obtain the ecological status of mangroves from remote sensed imagery [4,5,7,9,10]. However, high-resolution optical remote sensed images are not as readily accessible as public medium-resolution remote sensed images [11,12,13] (e.g., LandSat series and Synthetic Aperture Radar from Sentinel Satellite) [4].

Furthermore, optical remote sensed images are susceptible to interference from cloud and precipitation, which further reduces the number of available optical images [14]. Drone imagery data can effectively address this limitation, as drones can fly below the cloud cover, obtaining ultra-high-resolution images without being affected by cloudy weather conditions [15,16]. Additionally, the multispectral frequency band information provided by satellite images is frequently employed to differentiate the characteristics of disparate ground objects [17,18]. Nevertheless, despite the availability of various spectral bands in satellite image data, the low resolution of such images still presents a challenge in distinguishing between mangrove species [19]. To address this issue, we used multispectral sensors mounted on drones to capture multi-spectral, high-resolution images encompassing visible light bands, red-edge band, and near-infrared band [19,20].

Regarding the mangrove extraction algorithm, research has been conducted on utilizing a mangrove index based on remote sensed images to identify and obtain an accurate delineation of the mangrove range [21]. The method of mangrove information extraction based on spectral index has been demonstrated to offer high efficiency and broad applicability. In addition to investigating the potential applications of conventional spectral indices in mangrove information extraction, researchers have also attempted to construct mangrove indices based on the distinctive spectral properties of mangrove vegetation [22,23]. Shi et al. (2016) [24] proposed the Normalized Difference Mangrove Index (NDMI) based on Landsat-8 OLI images as a means of enhancing the separability of mangrove and terrestrial vegetation. Diniz et al. (2019) [25] proposed the Modular Mangrove Recognition Index (MMRI) based on Landsat TOA images for the analysis of dynamic changes in Brazilian mangroves from 1985 to 2018. Huete et al. (2002) [26] combined Landsat-5 TM, Landsat-7 ETM+ and Landsat-8 OLI images to construct and combined the Enhanced Vegetation Index (EVI), Modified Soil-Adjusted Vegetation Index (MSA-VI) [27], Normalized Difference Vegetation Index (NDVI) [28] and NDMI to analyze the health of mangrove forests in a coastal lagoon.

The methods above are relatively straightforward to implement. However, the robustness of these methods is still being determined, given the necessity for applying diverse thresholds [29,30] in the context of images captured under disparate conditions, including those about time and sun angle. To address this challenge, data-driven machine learning offers an alternative approach [31]. For instance, researchers have employed machine learning algorithms, such as Random Forest (RF) [32], Support Vector Machine (SVM) [33], and Artificial Neural Networks (ANNs) [34], to extract mangroves from remote sensed images. However, the efficacy of machine learning based algorithms hinges on expertise, and the performance of the corresponding methodology is contingent on the quality of the handcrafted feature selection [31].

Deep learning algorithms have demonstrated remarkable efficacy in many remote sensed interpretation and image segmentation scenarios [35]. The extensive utilization of deep learning algorithms in remote sensed interpretation has led to their emergence as a superior alternative to traditional remote sensed algorithms and conventional machine learning algorithms. It can autonomously learn high-level semantic information in images, which enhances segmentation accuracy. This capability has led to a growing use of deep learning algorithms in remote sensed image processing, particularly for identifying and classifying mangrove ecosystems [36]. In a study conducted by Dang et al. (2020) [37], the performance of the U-Net model and SVM in classifying mangrove forests was compared. The findings revealed that the U-Net model demonstrated superior performance, achieving an accuracy rate of 90% and a Kappa coefficient value of 0.84. In comparison, the SVM attained an accuracy of 86% and a Kappa coefficient of 0.78 [37]. Faza et al. (2018) [38] built a deep neural network architecture and analyzed it for mangrove sprout plant classification. The result from the architectural methodology of this research reached the lowest training error of 0.1345 and the highest testing accuracy value of 98% [38].

The following section outlines the structure of the paper. Section 2 presents an overview of the study region and the datasets. Section 3 provides a detailed account of the architectural design of the MangroveNet and AttCloudNet+, the data processing methodology, the extraction of mangrove areas, and the identification of mangrove species using our deep learning models. Section 4 presents the precision indices performance and model validation of the MangroveNet and other CNN segmentation models. Furthermore, the AttCloudNet+ is used to categorize mangrove species and evaluate their performance compared to other traditional methods and machine learning algorithms (e.g., ISODATA, Kmeans [39], RF, SVM, etc.). In the discussion section, factors affecting the results of each model, the effect of shadow cover on the mangrove extraction misclassification in drone images, and the limitations of our models are demonstrated. The conclusion is described in Section 6. This study is highlighted by developing modified deep-learning architectures to study mangroves and their species identification via drone multispectral high-resolution imagery.

## 2. Study Area and Data

### 2.1. Study Area

The selected research area is situated within the Guangdong Zhanjiang Mangrove National Nature Reserve’s largest mangrove national nature reserve in China [40]. The reserve is geo-located on the Leizhou Peninsula of Guangdong Province, occupying a geographically distinctive position at the southernmost extremity of the Chinese mainland [41]. The coordinates of this region are between 109°40′ and 110°35′ east longitude and 20°14′ and 21°35′ north latitude [41]. The remainder of the area is characterized by terraces and plains with gentle slopes [42]. The highest altitude of the peninsula is 382 m, and the region is devoid of elevated mountainous terrain. The mean annual temperature of the reserve is approximately 23 °C, with a maximum temperature of 38.8 °C and a minimum temperature of −1.4 °C [43]. The average annual precipitation in the area is 1534.6 mm, indicating a clear distinction between wet and dry seasons.

This study acquired drone image data near Jinniu Island in the Huguang Jinxing partition, situated on the eastern Zhanjiang Mangrove National Nature Reserve, as shown in Figure 1. The ecosystem in this region is relatively complete and possesses high protection and scientific research value [44]. A comprehensive scientific investigation report and research in this reserve revealed the existence of nine basic mangrove community types within the reserve [45]. These include the *Avicennia marina*, the *Rhizophora stylosa*, the *Aegiceras corniculatum*, the *Sonneratia apetala*, the *Kandelia obovata*, the *Bruguiera gymnorrhiza*, the *Acrostichum aureum*, the *Acanthus ilicifolius*, as well as the *Excoecaria agallocha* [46]. The most prevalent mangrove species near Jinniu Island are *Rhizophora stylosa*, *Aegiceras corniculatum*, *Sonneratia apetala*, and *Avicennia marina*.

### 2.2. Imagery Data from DJI Drones

This study employed a DJI Phantom 4RTK multispectral drone to capture the initial data collection phase in October 2023 for the mangrove area in Jinniu island. The drone is equipped with an all-in-one multispectral imaging system that integrates a visible band and five monochromatic bands [47], including blue, green, red, red-edge, and near-infrared (see Table 1 for details of the spectral ranges of each band). The sensor lenses in these bands are integrated on the same sensor platform (Figure 2) and then loaded on the DJI Phantom 4RTK multispectral drone [47].

This system is capable of fulfilling the tasks of both visible and multispectral images [48]. This type of drone is equipped with an RTK (Real-Time Kinematic, real-time Dynamic Carrier Phase Difference technology) positioning module, which offers a position accuracy of 1 cm ± 1 ppm horizontally and 1.5 cm ± 1 ppm vertically [49]. One ppm represents an error increase of 1 mm for every 1 km the aircraft travels [49]. Consequently, this equipment can meet the required position accuracy standards without establishing ground control points.

A total of 3726 multispectral images were acquired in this data acquisition work, with the flight altitude set at 100 m, the side overlap degree set at 70%, and the forward overlap degree set at 80%. The elevation angle of the sensor was set at −90 degrees, the photo mode was equal time interval, the flight speed was 10.2 m/s, and the ground sample distance (GSD) was 4.61 cm/pixel in ortho mode. The resolution of the drone image is considerably higher than that of the commercial satellite image, which has the highest resolution (0.15 m) and significant potential for identifying and classifying the spatial distribution of specific mangrove species. These drone imageries show that different mangrove species exhibit distinctive characteristics. The *Rhizophora stylosa* species is presented by a nearly circular dark green feature in the image. The *Aegiceras corniculatum*-*Avicennia marina* is also depicted as a green feature, albeit with a slightly lighter hue than that of the *Rhizophora stylosa*. Additionally, the area of the *Aegiceras corniculatum*-*Avicennia marina* is relatively limited, exhibiting a scattered distribution. In the image, the *Sonneratia apetala* presents a lighter green feature, and the edge features are more conspicuous (Figure 3).

## 3. Methodology

### 3.1. Deep Neural Network Model Design

#### 3.1.1. Application of Attention Mechanisms in Deep Learning

The attention mechanism [50] was integrated into the model structure to enhance the capability of extracting mangrove areas. The attention mechanism is a simple and effective module for convolutional neural networks dealing with 2D image data [50]. This module can obtain feature layers with channel-distributed weights and spatial-distributed weights from the channel and spatial dimensions of images, respectively. These two feature layers with weight information were then multiplied by the corresponding initial feature layers to complete the attention to the channel or spatial features. Among them, the computational process of the channel attention module is as follows:(1)$$W_{c}(F_{\mathrm{prev}})=\delta \left(\mathrm{MLP}\left(\mathrm{AvgPool}\left(F_{\mathrm{prev}}\right)\right)\mathrm{MLP}\left(\mathrm{MaxPool}\left(F_{\mathrm{prev}}\right)\right)\right)$$
where $F_{prev}$ is the result of the encoding process of the neural network model; *MLP* is a multilayer perceptron with shared weights; *AvgPool* is the average pooling operation; and *MaxPool* is the maximum pooling operation.

As the input data of the channel attention module, the size of the result $F_{prev}$ of the coding process was H×W×C. Then, global average pooling and global max pooling were performed to obtain two semantic feature layers, the size of which was 1×1×C. These features were then fed into a weighted multi-layer perceptron, and *ReLU* was used as the activation function to enhance the non-linear relationship between the feature layers. To obtain the weight coefficients of the different channels, the *Sigmoid* was used. Finally, a new feature layer based on channel attention could be obtained by multiplying the obtained channel weight coefficient with the original input feature layer.

The processing of the spatial attention mechanism is similar to the channel attention mechanism for an input feature layer of size H×W×C. First, average pooling and max pooling were performed along the channel direction to obtain a feature layer with the same number of layers and channels with spatial feature information [50]. Then, the results of average pooling and maximum pooling were concatenated. Then, a 7×7 convolution kernel was used for the convolution calculation, and the spatial feature weight coefficients were obtained after processing with a sigmoid activation function. Finally, the original input feature layer was multiplied by the spatial feature weight coefficients to obtain the new feature layer based on the spatial attention mechanism.(2)$$W_{s}(F_{\mathrm{prev}})=\delta \left({f\;}^{n\times n}\left[\mathrm{AvgPool}\left(F_{\mathrm{prev}}\right);\mathrm{MaxPool}\left(F_{\mathrm{prev}}\right)\right]\right)$$
where $f^{n\times n}$ is an n×n convolution process. Finally, the new feature layer based on the channel attention mechanism and the new feature layer based on the spatial attention mechanism were linked and input to the subsequent decoder. We connect the two attention mechanisms in parallel to process the image in spatial and channel simultaneously, while CBAM [50] puts the channel attention module in front of the spatial attention module. When processing images, the spectral features are as important as spatial features, so we connected the two types of attention in parallel. As a result, the deep learning model can fully use the band information and spatial features of the remote sensed image to identify and extract the mangrove areas and species more accurately [51].

#### 3.1.2. Novel Model of Integrating Multi-Scale Spectral and Spatial Information for Detecting Mangrove Area

For effectively extracting mangrove area, we designed MangroveNet, a novel enhanced model with integrating multi-scale attentional visual field information. The overall model structure is shown in Figure 4. The model comprises three main components: an encoding structure, a corresponding decoding section, and a multi-scale attention extraction module. In terms of sampling mode, the model imitates the sampling mode of SegNet architecture [52]. In order to enhance the model more lightweight, particularly in scenarios where hardware memory is limited, only two downsamples of maximum pooling operations were implemented. Consequently, this results in the formation of three distinct scale-specific feature layer views. It is postulated that during the processes of convolution and semantic information extraction, there exist critical spectral and spatial information that necessitates consideration across diverse scales of vision. The attention mechanism module can then be utilized to assign significant weights to these focused spectral or spatial information. Subsequently, the feature layer containing the high-weight spectral information and the feature layer comprising the spatial information are connected, thereby yielding a feature layer that integrates the essential information from both spectral and spatial perspectives. In this paper, the focus is on the critical spectral and spatial characteristics of the mangrove area. The integrated spectral and spatial information feature layer ($X_{con\_att}^{l}$) can be obtained according to the following calculation formula:(3)$$X_{con\_att}^{l}=W_{c}^{l}\times X^{l}+W_{s}^{l}\times X^{l}$$
where $W_{c}^{l}$ is the weight from channel attention; $W_{s}^{l}$ is the weight from spatial attention; *l* represents the number of feature layers; $X^{l}$ represents feature map in *l*-th feature layer. Subsequently, the feature layer needs to be concatenated with the convolution feature layer that has not been subjected to maximum pooling. However, the size of the feature layer is the same as before the attention mechanism module. Consequently, it is necessary to upsample it prior to its connection with the convolution feature layer that has not been sampled by max pooling. It is important to note that if the feature layer has been maximized twice, the number of up-samples should also be doubled to ensure that the initial size can be restored. The feature information layer with weight information from each scale is fused with the initial feature layer and we can obtain $X_{con\_total}$:(4)$$X_{con\_total}=\sum_{i=1}^{n}\left(UpSamp(X_{con\_att}^{i})+X_{con\_total}^{i-1}\right)$$
where *n* indicates the number of times about max pooling operations; $X_{con\_total}^{i-1}$ represents the result of the last integration. Specifically, when *i* − 1 = 0, $X_{con\_total}^{0}$ represents the initial layer have been convolved but not max-pooled; *UpSamp* is the upsampling operations. In addition, the operations for low-resolution feature layers entail the retention of index information from each encoder maximum pooling operation, which is then employed in upsampling at the decoding. After integrating high-weight spectral and spatial information of all scales and normal convolution sampling information, the feature layers were fed into *Sigmoid* for segmentation [52], the formulation is indicated as follows:(5)$$X_{sigmoid}=\delta \left(\frac{1}{1+e^{-\left(X_{con\_total}+X_{upsampling}^{n}\right)}}\right)$$
where *n* is the number of upsampling operations; $X_{upsampling}^{n}$ represents the feature map after the *n*-th upsampling. In the training process, it is also necessary to compare the predicted result with the label value by using the cross-entropy loss function, and send the error back to the network for adjustment, until the predicted result meets the threshold for output [53]. The loss function ($L_{mangrove}$) can be calculated using the following formulation:(6)$$L_{mangrove}=-\frac{1}{n}\sum_{i=1}^{n}\left(X_{i}^{t}\log\left(X_{i}^{p}\right)+(1-X_{i}^{t})\log(1-X_{i}^{p})\right)$$
where *n* is the total pixel number of features; $X_{i}^{t}$ represents the true value and $X_{i}^{p}$ is the prediction value.

#### 3.1.3. Improved Model for Identifying Mangrove Species

Due to mangrove species identification is more complex than mangrove extraction, we choose a deeper neural network structure—CloudNet+ as the base model for species identification. CloudNet+ is a fully convolutional neural network model with a deep learning encoder–decoder structure [54], mainly used for cloud detection and segmentation in Landsat-8 optical imagery. The model has several encapsulated blocks, including Feedforward Block, Expanding Block, Contracting Block, and Upsampling Block, as shown in Figure 5. The Contracting Block contains three successive convolutional layers and a parallel 1×1 convolutional layer, and after both complete the convolutional operation, the results of each are summed and then maximally pooled [55]. The Feedforward Block joined and pooled the results obtained from the Contracting Block, and finally, the results of each branch were pooled and accumulated [55]. The input data of the Expanding Block came from the Contracting Block and the Feedforward Block. The Feedforward Block result was convolved twice, summed with the Contracting Block, and then fed into the Upsampling Block. The result of the Feedforward Block was convolved twice and summed with the Contracting Block. Finally, the Upsampling Block performed an upsampling operation on the incoming feature layer to restore the image to its original size. The backward propagation of these blocks was trained using the Filtered Jaccard Loss function, which avoided the misjudgment of predicted values caused by the absence of valid values in the labels. Following the success of CloudNet+ in segmenting cloud-covered areas in remote sensed images, this study builds on its foundations and improves it to identify mangrove species. To enhance the efficiency of the model using multispectral images, we have elected to augment the channels and spatial attention mechanisms between down sampling and upsampling. This approach is intended to take full advantage of the spectral and spatial information of the images, thereby forming an improved AttCloudNet+ model.

### 3.2. Data Pre-Processing

The limitations of the field of view and the high degree of overlap in drone photography mean that the small aerial drone images taken during the mission must first be stitched together. Due to the inevitable occurrence of some distortions during shooting, the overlapping regions of the images will produce a certain degree of bias during stitching, so it is necessary to eliminate some severe distortions or excessively unsmooth regions after stitching. In this study, the Agisoft Metashape 2.1.1 software [56] (https://www.agisoft.com/, accessed on 28 May 2024) was used to process the acquired high-resolution multispectral drone images into a dense point cloud, textured polygon models, DSMs and georeferenced true orthophotos. The acquired high-resolution multispectral drone images were mosaicked, and orthophotos of the study area were generated. The drone remote sensed images of the mangrove areas in each band were mosaicked with an average resolution of approximately 13,137 × 15,061 pixels. A total of 18 multi-phase sampling orthophotos were generated and stored in TIFF format.

Drone images are often affected by the shape of the Earth’s surface and map projections during acquisition and transmission. Therefore, geographic correction is required to ensure the accuracy and reliability of the geographic information. Geo-correction is a geometric correction method that realizes the geo-localization of images by establishing an accurate correlation [57] between the pixels of a drone remote sensed image and the geographic coordinate points on the Earth’s surface. This correction corrects for image deformation due to the Earth’s curvature, map projection, and surface undulation. The principle of geo-correction in this study includes the following steps:

(1) Selection of control points: Before geo-correction, a set of control points must be selected. In this study, the accuracy of the geo-correction process was ensured by uniformly selecting five different geographic feature points within the image of the mangrove area.

(2) Performing the correction: During the correction process, each image pixel was moved to an exact position in the geographic coordinate system through geometric transformations (e.g., polynomial transformation or cubic spline interpolation) based on the known control points.

In this study, the 16-bit multispectral drone orthoimages acquired after the above operations were converted to 8-bit orthoimages [58] using the Geospatial Data Abstraction Library (GDAL, https://www.gdal.org, accessed on 15 July 2024). Then, the geographic coordinate system of all images was converted to the Universal Transverse Mercator (UTM) projection coordinate system using GDAL. Within the mangrove area of Zhanjiang, the reference of the projected coordinate system of the image was set at WGS_1984_UTM_Zone_49N. Since the original image size is large for model inputting, each multispectral drone orthoimage was segmented into an image of 512 × 512 pixels to create the image training dataset for the deep neural network in this study. During the segmentation process, an overlap of 150 pixels was ensured between each image and its neighboring images, which ensured both a sufficient number of training datasets and utilized the edge feature information of the images effectively.

### 3.3. Identify Mangrove Areas and Species

Figure 6 shows the flowchart of this study. In this study, we used multispectral drone images with five bands as our research data. In the aspect of preprocessing, we performed radiometric correction, geographical correction, bit depth conversion, projection transformation, and image subset. MangroveNet was employed to extract the extent of mangrove areas from high-resolution multispectral drone imagery. AttCloudNet+ was employed to identify the spatial distribution of different species within the mangrove area. Multispectral images of mangroves and their surrounding areas on Jinniu Island were selected as training and validation samples for this study. In both tasks, the images were divided into training datasets, validation datasets, and prediction datasets. After data training and fine-tuning, the optimal parameters were obtained, and the prediction data were predicted. During the prediction step, the two models were applied to their respective corresponding data, namely high-resolution multispectral drone images of mangroves in the study area on Jinniu Island. Finally, the results of mangrove area extraction and species identification were obtained. In the post-processing step, we needed to mosaic the image patches, and calculate and analyze the area of the mangroves and their species.

#### 3.3.1. Training Dataset Generation

The multispectral drone imagery training data used in this study was collected from the DJI Phantom 4RTK multispectral sensor. The area chosen for generating the training dataset is located in and around the mangrove forest of Jinniu Island. We chose this region because the mangrove forests are lush and well-protected, making the boundary between the water and the mangrove areas more distinct. Nevertheless, we selected different methods for mangrove area extraction and mangrove species identification. For mangrove range extraction, 25 subranges were selected as training data in this study, as shown in Figure 7a.

An integrated mangrove region with concentrated and abundant mangrove species in this area was selected for species identification. The training dataset for the mangrove species identification task was also selected from the Jinniu Island mangrove forest, focusing on areas with a high concentration of mangrove species types. The total area of this region is 3.12 hectares. The field investigation revealed that the mangrove tree species prevalent in this region include *Rhizophora stylosa*, *Aegiceras corniculatum*, *Sonneratia apetala*, and *Avicennia marina*. In some areas, there is a notable species group between *Aegiceras corniculatum* and *Avicennia marina*, manifesting as *Aegiceras corniculatum*-*Avicennia marina* tree clusters (Figure 8).

Accordingly, the primary mangrove species identifications in this integrated mangrove region are the *Rhizophora stylosa*, the *Aegiceras corniculatum*-*Avicennia marina*, the *Sonneratia apetala*, and the non-mangrove ground objects (e.g., water bodies, aquaculture fishing rows, and roads), as shown in Figure 9.

To generate the label dataset [59], we first tried the commonly used annotation tool Labelme (graphical interface annotation software widely used for image annotation) [60]. However, it was found that the edges marked by Labelme were relatively simple, which could not meet our experimental requirements of accurate segmentation tasks. Therefore, in this study, different labeling strategies were used for the two parts of the research tasks concerning the characteristics of each band of the drone multispectral image. For the mangrove area extraction task, we used threshold extraction combined with visual interpretation to accurately annotate the multispectral images to achieve high accuracy, as shown in Figure 7b,c. First, the mangrove area was roughly identified using the exponential thresholding method, and then the misidentified areas were corrected by experienced personnel. It was challenging to perform automated labeling for the mangrove species identification task due to numerous classification categories. Therefore, this study adopted a direct manual labeling method for the mangrove species identification task and used expert experience to ensure the accuracy of these labels, as shown in Figure 9. As the area of the label was also the same as that of the image, each label also had to be segmented and cropped into 512 × 512 pixels. In the segmentation process, each image had to overlap 150 pixels with the neighboring images, as in the image segmentation process, to ensure that the label matched the image data.

#### 3.3.2. Training and Prediction

In this part, we first describe our experimental setting. The experimental platform of this study employs an Intel 11th Gen Intel^®^ Core™ i7-11700@2.50 GHz 2.50 GHz processor equipped with 32.0 GB of memory and an NVIDIA GeForce GTX 4060Ti graphics card. In the experiment, Anaconda3 (64-bit) was employed as the carrier for environment configuration. The Windows 10 Professional 64-bit operating system was utilized to create the Keras tool, which is integrated into the framework and is used to construct the model [61]. The CUDA 11.2.2 version corresponds to the computer configuration and is selected as the GPU computing platform, and cuDNN 8.9.2 is installed as the deep learning GPU acceleration library. Finally, PyCharm Community Edition 2021.3.3 is employed to develop and compile the deep learning program.

The training dataset was prepared using high-resolution multispectral drone remote sensed images in five bands: B, G, R, RE, and NIR. In the mangrove range extraction task, 29,808 segmented training data were obtained in each band after pre-processing. 23,848 segmented images (80% of all segmented images) were used to train MangroveNet, and the remaining 5960 sub-images were used as a validation dataset in the training process [62]. The mangrove species distribution prediction task obtained 10,560 segmented training data for each band. Among them, 8448 segmented images (80% of all segmented images) were used to train AttCloudNet+ [62], and the remaining 2112 sub-images were used as a validation dataset in the training process. In the training process, the learning rate was set to 0.0001, the number of training in the same batch (Batch size) was set to 4, and the maximum number of iterations (Maximun iteration) was set to 1000. After training this network, this study predicted the extent of mangrove forests and the spatial distribution status of their species in the prediction data collected from the mangrove area of Jinniu Island, in which the prediction data contained a total of 3120 segmented sub-images.

### 3.4. Accuracy Metrics

#### 3.4.1. Evaluation Criteria of Mangrove Range Extraction

In this study, five evaluation indicators were selected to assess the prediction results of each model, including Accuracy, F1-Score, Precision, Recall, and mIoU [63] (Mean Intersection over Union), as shown in Equations (7)–(11).(7)$$\mathrm{Accuracy}=\frac{\mathrm{TP}+\mathrm{TN}}{\mathrm{TP}+\mathrm{TN}+\mathrm{FP}+\mathrm{FN}}$$(8)$$\mathrm{Precision}=\frac{\mathrm{TP}}{\mathrm{TP}+\mathrm{FP}}$$(9)$$\mathrm{Recall}=\frac{\mathrm{TP}}{\mathrm{TP}+\mathrm{FN}}$$(10)$$F1-\mathrm{Score}=\frac{2\times \;\mathrm{Precision}\;\times \;\mathrm{Recall}}{\mathrm{Precision}+\mathrm{Recall}}$$(11)$$\mathrm{mIoU}=\frac{1}{2}\times (\frac{\mathrm{TP}}{\mathrm{TP}+\mathrm{FP}+\mathrm{FN}}+\frac{\mathrm{TN}}{\mathrm{TN}+\mathrm{FN}+\mathrm{FP}})$$

*TP*, *TN*, *FP*, and *FN* correspond to true positives, true negatives, false positives, and false negatives, respectively, in the confusion matrix of the prediction results [64], as shown in Figure 10a. *TP* is the correctly categorized mangrove pixels; *TN* is the correctly categorized non-mangrove pixels; *FP* is the incorrectly categorized non-mangrove pixels incorrectly identified as mangrove; and *FN* is the mangrove pixels incorrectly identified as non-mangrove. The confusion matrix could then be used to construct a variety of metrics to evaluate the results. Accuracy represents the proportion of correctly recognized pixels to the total number of pixels; Precision is defined as the proportion of correctly recognized mangrove pixels to the number of pixels recognized as mangrove; Recall represents the proportion of correctly recognized mangrove pixels to the number of real mangrove pixels in the labels; F1-Score reveals the weights summed average of Precision and Recall; and mIoU measures the degree of similarity of the model’s prediction results to the real labels, as shown in Figure 10b.

#### 3.4.2. Indicators of Mangrove Species Identification

Due to the considerable number of identification results in the mangrove species identification task, various evaluation strategies are employed to assess the accuracy of the results. The selected criteria for evaluating the identification accuracy of AttCloudNet+ and other traditional supervised classification, unsupervised classification, common deep learning algorithms, and machine learning algorithms are the Confusion Matrix, Kappa coefficient, and Overall Accuracy [65].

The Confusion Matrix, also known as the Error Matrix, is an array used to compare the number of image elements classified into a category and the number of ground tests for that category. Typically, the matrix columns represent the reference data, and the rows represent the category data obtained from the classification results of the remote sensed data. The Kappa coefficient is an index that can show more comprehensively the degree of consistency between the predicted results and the true labels (Table 2). The overall accuracy can indicate the general accuracy of the classification results without considering the situation of a specific category. The Kappa coefficient can solve this problem well and analyze the classification accuracy more comprehensively and objectively. The expression of the Kappa coefficient is as follows:(12)$$\mathrm{Kappa}=\frac{Accuracy-P_{e}}{1-P_{e}}$$(13)$$P_{e}=\frac{\left(TP+FN\right)\times (FP+TN)\times (TP+FP)\times (FN+TN)}{{\left(TP+FP+FN+TN\right)}^{2}}$$
where Accuracy represents the observed agreement probability; $P_{e}$ is the expected agreement probability, which represents the expected observed agreement probability in the case of randomness. The explanation of *TP*, *TN*, *FP*, and *FN* are shown in Figure 10a.

The Overall Accuracy (OA) equals the sum of correctly classified pixels divided by the total number of pixels. The number of correctly classified pixels is distributed along the diagonal of the confusion matrix, and the total number of pixels is equal to the total number of pixels from all actual reference sources. The overall classification accuracy is the probability that, for each random sample, the classified result matches the test data type.

## 4. Results

In consideration of the spatial and multispectral characteristics inherent to high-resolution drone remote sensed imagery, MangroveNet and AttCloudNet+ were employed to detect the extent of mangrove forests and identify the spatial distribution of mangrove species. A comparison was conducted between the initial results obtained from our models and other mainstream semantic segmentation models, including UNet [67], SegNet [52], SegFormer [68], and other UNet-based variant neural network models (e.g., DeepUNet [69], ResUNet [70]). Moreover, results under other traditional supervised or unsupervised algorithms were also compared with ours. The quantitative comparison results of the area of interest in the mangrove reserve region are demonstrated in Table 3 and Figure 11.

### 4.1. Mangrove Area Extraction and Evaluation

Table 3 reveals the performance of other diverse segmentation models and our promoted model on the evaluation criteria. Table 3 is based on five metrics—Accuracy, F1-score, mIoU, Precision, and Recall. We obtained the following conclusions from Table 3: (1) The MangroveNet model exhibits optimal performance in Accuracy, F1-Score, mIoU, and Precision. Regarding the recall indicator, AttCloudNet+ demonstrated the most favorable performance. (2) The distinctive spatial and spectral characteristics of ground objects in remote sensed images present a challenge for the conventional deep learning model based on convolution and pooling layers, which cannot adequately attend to and enhance mangroves’ spatial and spectral characteristics in remote sensed images. Consequently, models other than MangroveNet perform poorly in most indicators. (3) Similarly, although AttCloudNet+ is equipped with spatial and spectral attention mechanisms, its considerable model size hinders its performance in mangrove-non-mangrove binary classification. The subsequent mangrove species segmentation experiments also demonstrated that AttCloudNet+ is better suited to mangrove species’ multi-category identification tasks. (4) SegFormer model is an advanced Transformer framework for semantic segmentation, with a hierarchical Transformer encoder. In our experiments, the model outperforms UNet and DeepUNet. However, since the data processed in this experiment is multi-spectral remote sensing data, the performance of SegFormer may be different from that in ordinary test images. Furthermore, we focused on comparing the fluctuations in Loss and mIoU across a range of deep learning models, including MangroveNet, AttCloudNet+, DeepUNet, and SegNet, about the duration of the training process. Following over 35 training rounds for all models, both Loss and mIoU demonstrated a stable state. Compared to other models, MangroveNet exhibited the lowest Loss value, while the mIoU index demonstrated the highest rate of increase and reached the highest value after 20 training rounds (Appendix A).

Each model’s results were visualized to facilitate comparison between the models and ascertain the effect of each model (Figure 11). A comparative analysis showed that the prediction results of MangroveNet were the most closely with the reality. Conversely, the prediction results of alternative models exhibited varying degrees of misidentification. From the results of each model after visualization, we can compare and find the following: (1) While creating the labels, the water surface within the mangrove gap was not identified. However, it is important to note that the mangrove community is not a single entity; it is distributed with water surfaces in its gap. The UNet, DeepUNet, ResUNet, and SegNet models could not accurately identify the water surface within the mangrove gap. In contrast, MangroveNet and AttCloudNet+ demonstrated superior performance, with MangroveNet offering a more precise reflection of the actual ground situation. This outcome suggests that incorporating spatial and spectral channel attention mechanisms can enhance recognition accuracy. (2) The UNet model erroneously identified a bare road as a mangrove area, while the AttResUNet only partially excluded the aquaculture fisheries, erroneously identifying them as mangroves. Furthermore, the edge of the mangrove forest was identified with considerable ambiguity in the recognition results of ResUNet. Based on the recognition outcomes of these models, ResUNet exhibited the least effective recognition performance. MangroveNet represents the optimal model, offering a comprehensive and intuitive approach to performance evaluation and a robust quantitative assessment of indicators.

### 4.2. Mangrove Species Identification and Evaluation

Furthermore, the predictions of AttCloudNet+ are presented (Figure 12) and compared with those of other traditional supervised and unsupervised models, machine learning models, and deep learning models (Figure 13). The results demonstrate that the AttCloudNet+ prediction is the most accurate, while SegNet exhibits superior performance in predicting the maturation of *Sonneratia apetala* but inferior performance in predicting *Aegiceras corniculatum* and *Rhizophora stylosa*. The SVM demonstrates a robust prediction efficacy for the *Rhizophora stylosa* and *Aegiceras corniculatum* yet exhibits a comparatively diminished prediction efficacy for the *Sonneratia apetala*. Additionally, RF demonstrated a notable predictive capability for *Rhizophora stylosa* and *Aegiceras corniculatum*. However, it is noteworthy that this model exhibited a certain degree of inaccuracy in identifying the *Sonneratia apetala*, and its capacity to eliminate the influence of aquaculture fishing rows needed to be improved. K-means can identify the range of the *Aegiceras corniculatum* to a certain extent; however, it cannot accurately identify the range of the other two mangroves and cannot eliminate the influence of aquaculture fishing rows. ISODATA [71] misidentified the *Rhizophora stylosa* as the *Sonneratia apetala*, the *Aegiceras corniculatum* as the *Rhizophora stylosa*, and the *Sonneratia apetala* as the *Aegiceras corniculatum*, and it was unable to eliminate the influence of aquaculture fishing rows.

We calculated the confusion matrix of mangrove major species categories. The confusion matrix of species identification results (Figure 14) in this study demonstrated that following the implementation of the enhanced AttCloudNet+ for predicting potential species within drone images, the classification accuracy (the maximum value is 1) of various categories prediction results is as follows: *Aegiceras corniculatum*-*Avicennia marina* (0.86), *Rhizophora stylosa* (0.86), *Sonneratia apetala* (0.72), and other objects (0.90). The primary cause of the misidentification of *Rhizophora stylosa* is the erroneous classification of 11% of *Aegiceras corniculatum*-*Avicennia marina* as *Rhizophora stylosa*. This may be attributed to the fact that the color and spectral characteristics of *Aegiceras corniculatum*-*Avicennia marina* and *Rhizophora stylosa* exhibit a similar spatial distribution, which may result in the misidentification of the model. The model’s prediction of *Sonneratia apetala* exhibited the lowest accuracy, primarily due to the misidentification of 19% of *Sonneratia apetala* as *Rhizophora stylosa*. This discrepancy may be attributed to the limited availability of *Sonneratia apetala* samples in the sampling area, which impairs the model’s ability to discern the characteristics of *Sonneratia apetala* during training effectively. Additionally, the similarity in morphology between *Sonneratia apetala* and other nearby mangrove species may contribute to the confusion.

Table 4 presents a comparative analysis of the AttCloudNet+ with traditional supervised classification, unsupervised classification, machine learning, and deep learning models. The effectiveness of each method in identifying mangrove species was evaluated by Kappa coefficient and the overall accuracy (OA) index. Table 4 demonstrated that the mangrove species identification prediction results using AttCloudNet+ exhibited the optimal performance in terms of the Kappa coefficient and the OA index. ISODATA has the worst Kappa coefficient and OA performance because the difference between mangrove species could be more apparent. For ISODATA, similar categories are often merged, leading to classification results that differ significantly from reality. Compared to traditional supervised or unsupervised classification algorithms, machine learning and deep learning algorithms show high Kappa coefficient and OA results, showing that machine learning and deep learning algorithms have great advantages in multi-category segmentation tasks of remote sensed images.

### 4.3. Calculation of Distribution Area

This study calculated the area of mangroves and their species using the area range deep learning model predicted.

i.To calculate the mangrove range, the grid mosaic calculation of the mangrove range prediction results is first performed. This is followed by the use of the TabulateArea function in the ArcPy Python API (Python 3.7 or later is recommended; ArcPy is a Python library for ArcGIS Desktop and ArcGIS Pro), which is employed to determine the distribution of pixels classified as mangroves within the specified area. Subsequently, the mangrove range area S is statistically calculated according to the pixel area size (Equation (14)), where *i* is the number of pixels, *r* is the ground resolution represented by pixels, and *N* is the total number of pixels within the target range.


(14)
$$S\;=\sum_{i=1}^{N}r_{i}^{2}$$


ii.About the distribution area of the mangrove species, it is evident that the mangrove species included in the predicted results have their respective result values (Figure 12). Thus, pixel statistics for each mangrove category must be conducted using the values resulting from the above. By calculating the number of pixels in a specific area and the area of a single pixel, it is possible to obtain the spatial distribution area of each mangrove species within the area. The mangrove species range area *S_j_* was calculated statistically (Equation (15)), where *i* is the number of pixels corresponding with the mangrove species, *j* is the species’ category represented by the gray value of the different prediction results, *r* is the ground resolution represented by pixels, and *N* is the total number of pixels within the target range.


(15)
$$S_{j}\;=\sum_{i=1}^{N}r_{i}^{2}$$


Once the spatial distribution of mangrove groups had been obtained, a detailed area distribution calculation was carried out to explore the specific quantitative parameters of mangrove group distribution. This was performed for various group distribution conditions. In the prediction process, the gray values of the feature markers representing various group categories were assigned the following labels: *Aegiceras corniculatum*-*Avicennia marina* (we set the gray value to 0), *Rhizophora stylosa* (gray value = 1), *Sonneratia apetala* (gray value = 2), and other features (gray value = 3). Accordingly, to ascertain the distribution area of the various mangrove groups in the prediction results, the number of gray values corresponding to each group can be subjected to statistical analysis by the partition. This will enable the distribution area of the various groups in the final recognition result to be determined according to the pixel size. Due to the lack of published mangrove species distribution dataset in this region and to verify the accuracy of the mangrove species identification model for the distribution area of various groups, this study manually identified and calculated the distribution area of the groups above within the study area. The resulting calculation and comparison are presented in Table 5. The accuracy of species areas (*A_sa_*) calculated by the model was defined by the areas of various mangrove groups calculated by the model and the areas of various mangrove groups depicted artificially. The *A_PA_* was calculated statistically (Equation (16)), where *i* is a serial number representing a mangrove species, *S_Di_* is the area calculated by manual delineation (Table 5), *S_Mi_* is the area calculated by AttCloudNet+ (Table 5), and *N* is the total number of mangrove species. The resulting value from this calculation was 87.66%, indicating an average accuracy of the mangrove species area calculation.(16)$$A_{sa}\;=\frac{1}{N}\sum_{i=1}^{N}\frac{\left|S_{D_{i}}-S_{M_{i}}\right|}{S_{D_{i}}}$$

## 5. Discussion of Issues

### 5.1. Factors Affecting the Results of Each Model

In the extraction of mangrove areas, this study showed low F1-score, Precision, Recall, and mIoU metrics when using other models (e.g., UNet, DeepUNet, ResUNet, AttCloudNet+, SegNet, etc.) for the mangrove area identification extraction, as shown in Table 3. These models showed low efficiency, which may be affected by the following reasons. UNet had the lowest metric score. Due to the simple structure of the model, it may not be able to handle the complex texture features of high-resolution drone images. DeepUNet is a variant of UNet that considers the image’s spectral features. However, the method ignored the spatial feature information, leading to uncertainty in identifying mangrove areas. SegNet has similar limitations to DeepUNet. Since ResUNet only considers spatial features and ignores spectral features, it may not perform as well as MangroveNet. Although some models performed poorly in mangrove area recognition, each model still had its own advantages and disadvantages. We sorted out the advantages and disadvantages of the deep learning models involved in our experiment, as shown in Table 6.

### 5.2. Impact of Shadows on Model Predictions

While extracting mangrove area identification, it was found that the results of mangrove area extraction were not satisfactory in some areas. Some areas were affected by the shadow of the mangrove projections in the light, resulting in these two types of features having close gray values and texture features in the image. This caused the model to incorrectly recognize the shadow of the mangrove forest in the light as part of the mangrove area in the prediction process, making the actual prediction result of the mangrove forest large.

This is because the model used in this study was temporarily unable to distinguish mangrove light shadows from normal mangrove areas. In future studies, mangrove image data with light shadows can be tried as negative samples to correct the model’s misrecognition of shaded areas in the training data. In addition, linear stretching of the raw multispectral drone image data can mitigate the effect of mangrove light shadows on the prediction process.

### 5.3. Limitations of the Current Model

Although our model has made some progress in the identification of mangrove species, when there are mixed tree species, our model finds it really difficult to define the respective ranges of mixed tree species, mainly because the phenomenon of mangrove species association causes difficulties in remote sensing image interpretation. In addition, mangrove seedlings are too small to be reflected in remote sensing images or drone images, so the accuracy of their recognition also needs to be improved.

## 6. Conclusions

In this study, we proposed enhanced deep learning models, namely MangroveNet and AttCloudNet+, to facilitate the detection of changes in the extent of mangrove habitats in the study area. This was achieved by identifying and extracting the salient features present in high-resolution multispectral drone spatial imagery obtained from the mangrove area of Jinniu Island in Zhanjiang Mangrove National Nature Reserve and the spatial distribution of mangrove species within the aforementioned study area. The accuracy of the validation results show that the MangroveNet model used in this study has a high degree of accuracy in identifying and extracting mangrove areas. Compared to other mainstream models, the accuracy of MangroveNet in the prediction process was improved to 99.13%, while the mIoU was improved to 98.11%. Moreover, our MangroveNet model shows the most optimal performance across the metrics: Accuracy, mIoU, Precision, and F1_Score.

The confusion matrix revealed that the classification accuracy of the various species prediction results reached 0.86 (the maximum value is 1) for *Aegiceras corniculatum*-*Avicennia marina*, 0.86 for *Rhizophora stylosa*, 0.72 for *Sonneratia apetala*, and 0.90 for other features, respectively, following the application of AttCloudNet+ to predict the possible mangrove species categories. Moreover, a comparison was conducted between the Kappa coefficient and overall accuracy (OA) of AttCloudNet+ and those of traditional supervised classification, unsupervised classification, and machine learning methods for species category prediction. The results demonstrate that the Kappa coefficient of AttCloudNet+ is 0.81, indicating that the model’s predictions for mangrove species are highly consistent with the actual situation. Furthermore, the overall classification accuracy of AttCloudNet+ reaches 0.87, which is the optimal performance among the six traditional supervised classification, unsupervised classification, machine learning, and other methods.

In conclusion, the MangroveNet and AttCloudNet+ proposed in this paper can efficiently and accurately obtain mangrove ranges and detect changes based on high-resolution multispectral drone spatial image data. Moreover, they can accurately predict and analyze the spatial distribution status of mangrove species. This approach offers an enhanced spatial and temporal resolution of mangrove forests and their species, providing a robust foundation for evaluating mangrove protection, monitoring, and assessing tropical and subtropical forest distribution. It offers scientific and technological support for assessing and monitoring mangrove protection effectiveness.

## Figures and Tables

**Figure 1 sensors-25-02540-f001:**
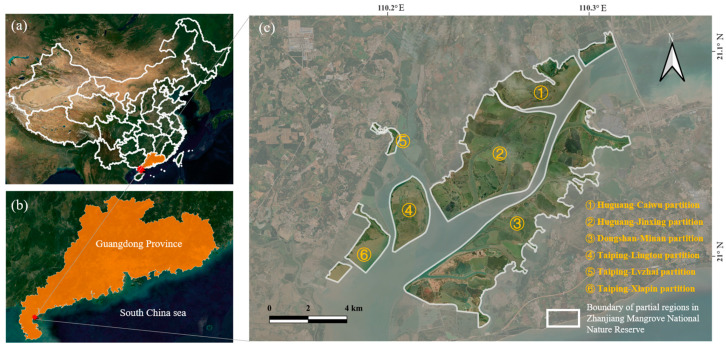
(**a**) The location of mangrove study area in China. The orange area is the Guangdong province, and the red pentagram represents the study area; (**b**) zoom in the study area; (**c**) the partial scope and brief info of Zhanjiang mangrove national nature reserve where the study area is located.

**Figure 2 sensors-25-02540-f002:**
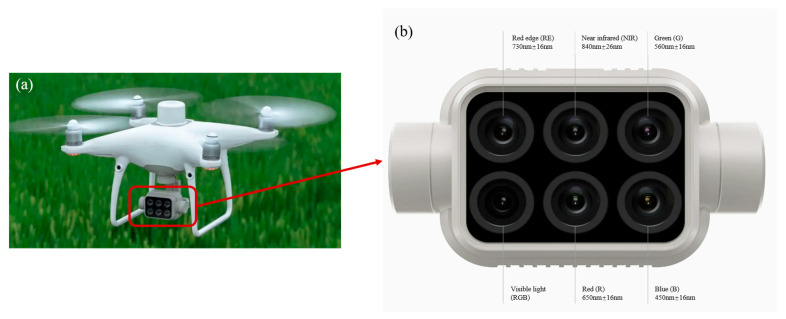
Schematic diagram of the DJI Phantom 4RTK multispectral sensor. (**a**) Appearance of DJI Phantom 4RTK; (**b**) multi-spectral sensor platform structure of DJI Phantom 4RTK.

**Figure 3 sensors-25-02540-f003:**
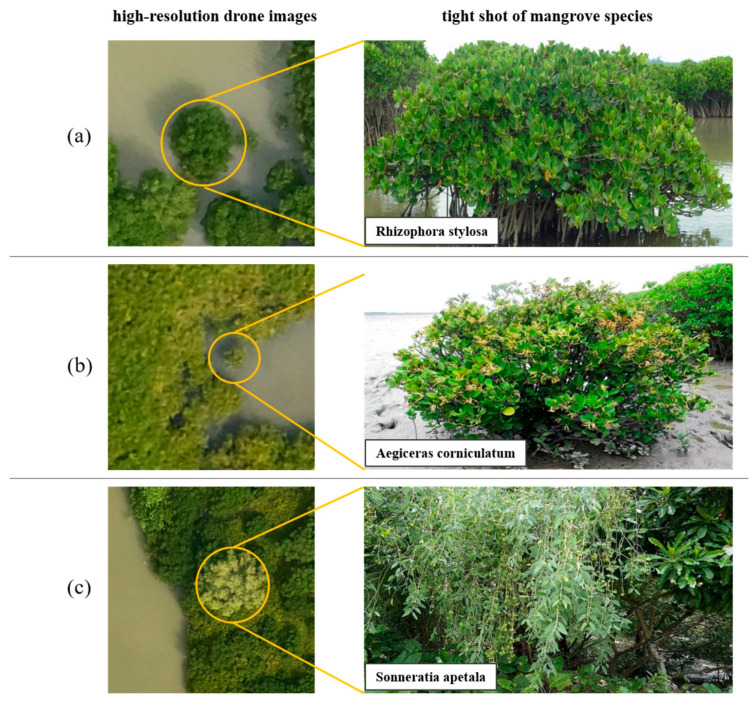
Features of different mangrove species in high-resolution drone images. (**a**) *Rhizophora stylosa* in drone image (**left**) and close-up photograph (**right**); (**b**,**c**) similar image cases of *Aegiceras corniculatum* and *Sonneratia apetala*.

**Figure 4 sensors-25-02540-f004:**
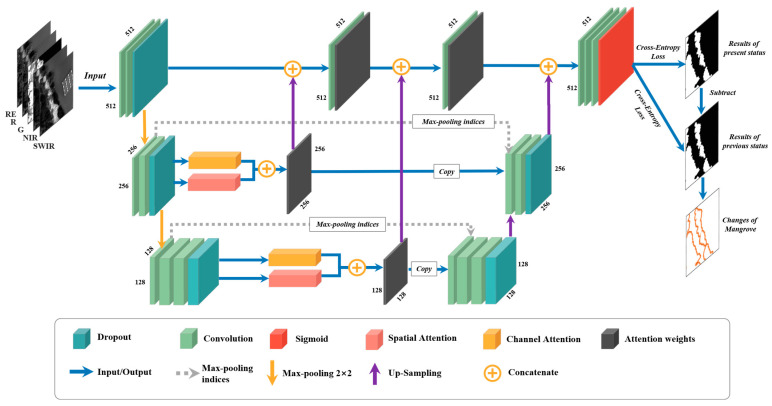
The internal structure of the MangroveNet model. Note that the numbers marked next to each convolution layer represent the pixel size of the layer.

**Figure 5 sensors-25-02540-f005:**
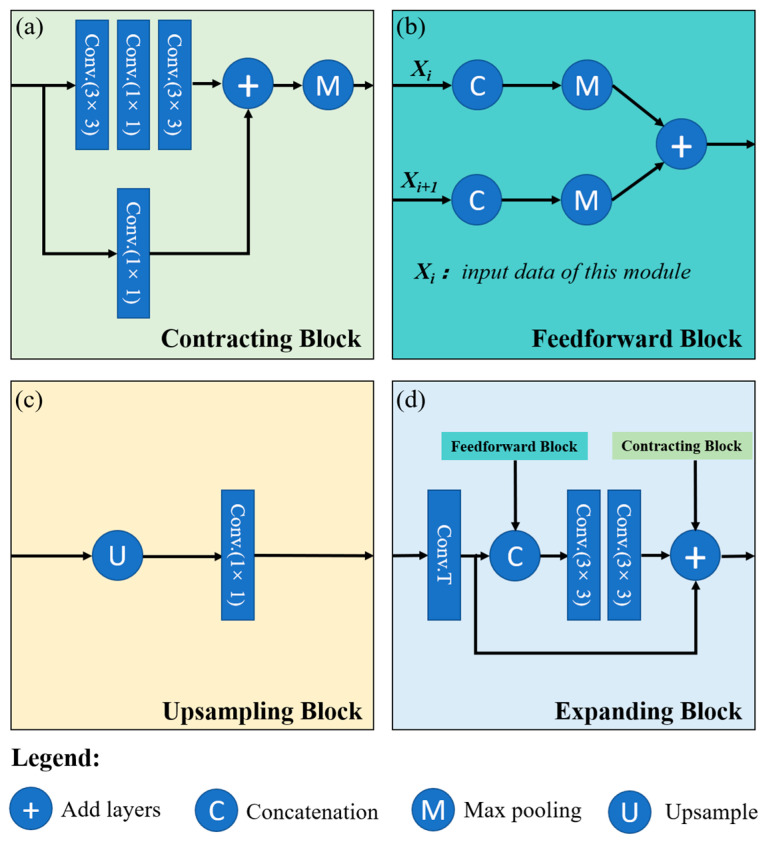
Schematic diagram of the internal structure in the base model [55]. (**a**) Contracting Block; (**b**) Feedforward Block; (**c**) Upsampling Block; (**d**) Expanding Block.

**Figure 6 sensors-25-02540-f006:**
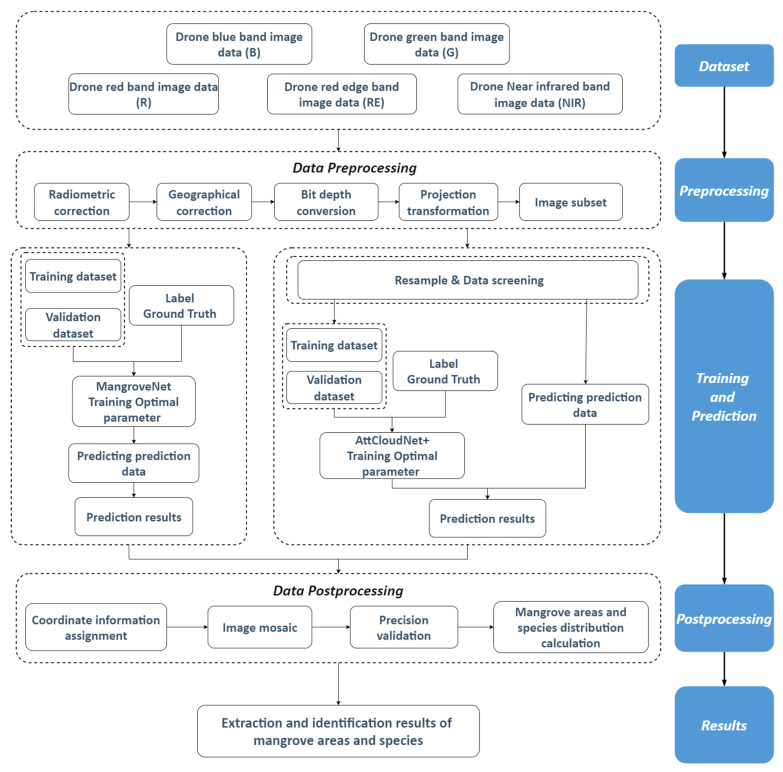
Flowchart of mangrove extraction and species identification using drone imagery.

**Figure 7 sensors-25-02540-f007:**
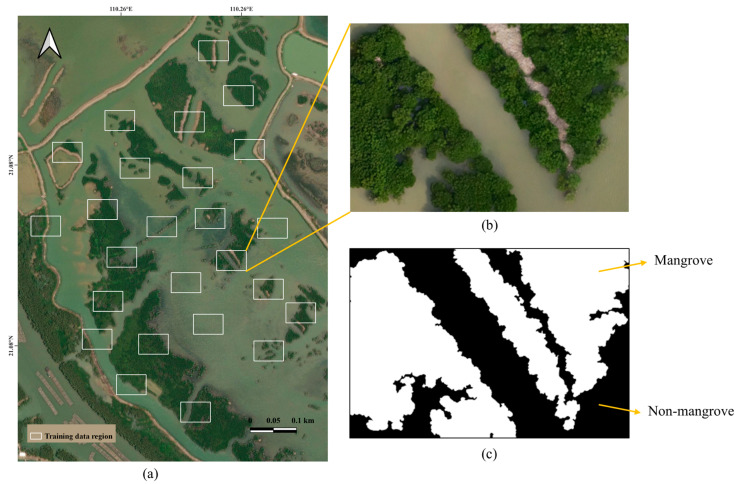
(**a**) Training dataset region for mangrove areas extraction; (**b**) example of a specifically selected training range; (**c**) label of the corresponding sample training range.

**Figure 8 sensors-25-02540-f008:**
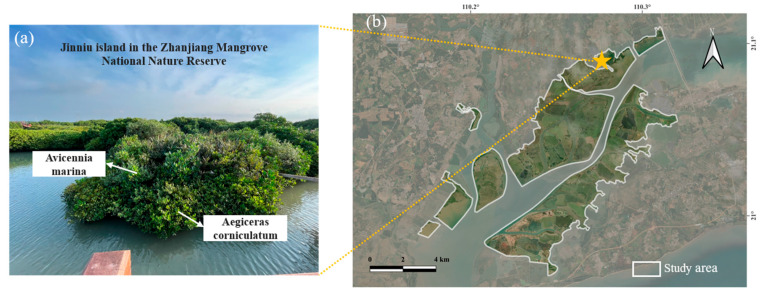
(**a**) In situ photos of the associated growth of *Aegiceras corniculatum* and *Avicennia marina*; (**b**) scope of the study area and the yellow pentagram represents the sampling position.

**Figure 9 sensors-25-02540-f009:**
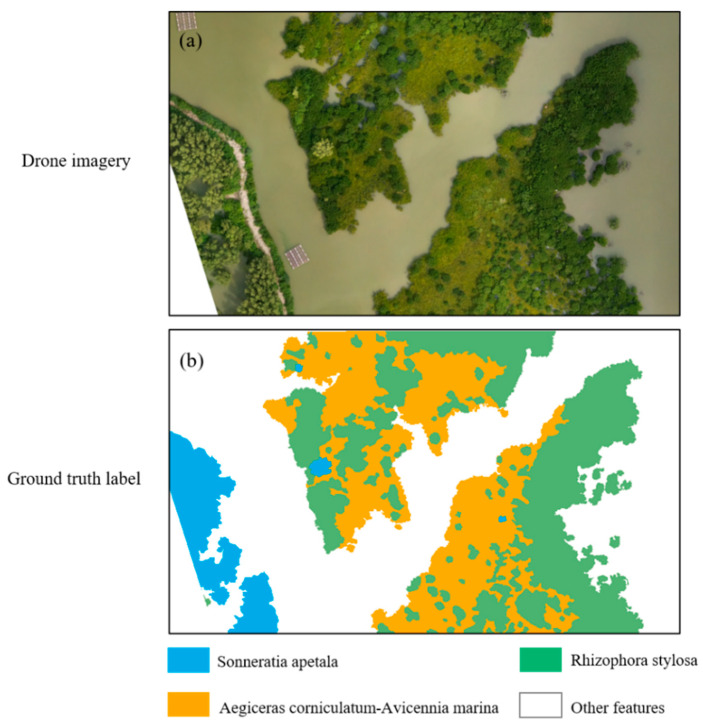
Training dataset region for mangrove species identification. (**a**) Drone in situ image of the selected mangrove region; (**b**) the corresponding mangrove species type label for this specific region.

**Figure 10 sensors-25-02540-f010:**
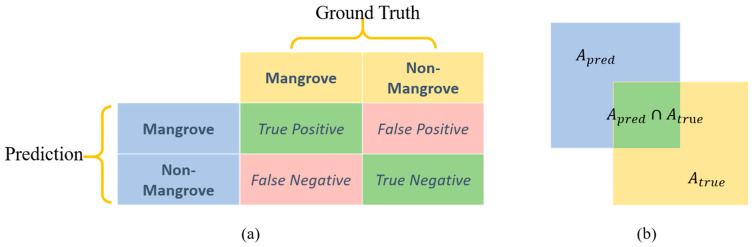
(**a**) Schematic of confusion matrix; (**b**) schematic of mIoU calculation process.

**Figure 11 sensors-25-02540-f011:**
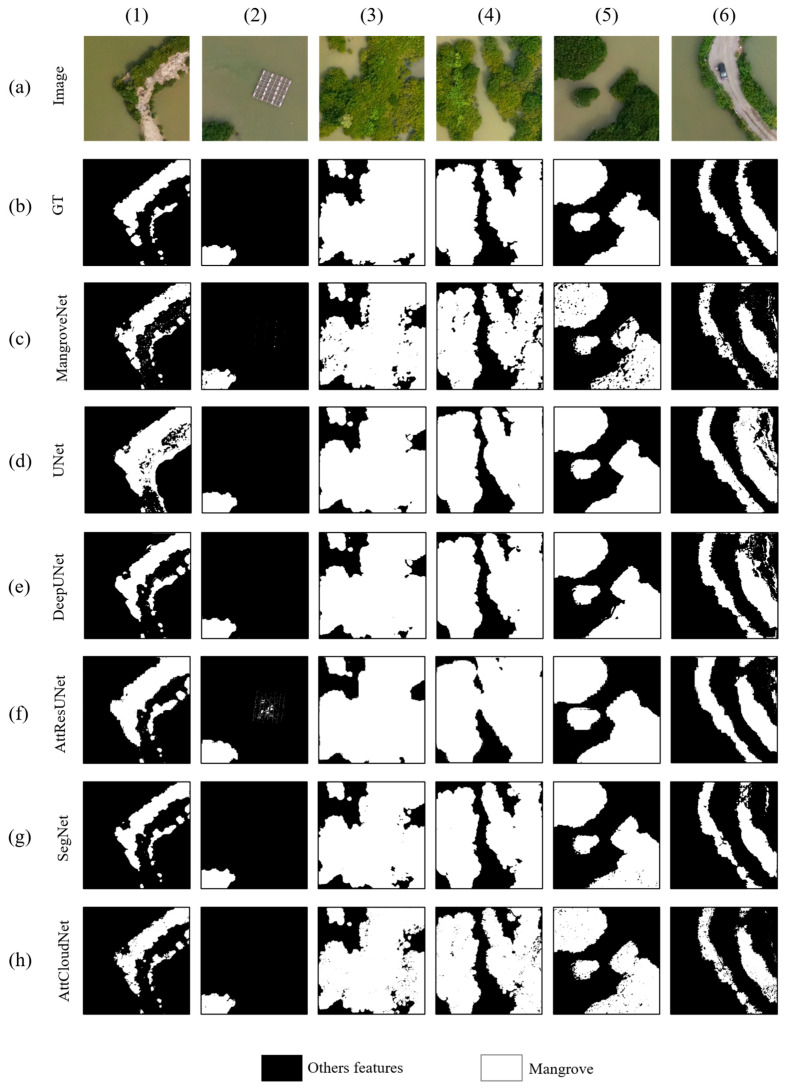
Comparison of different models’ prediction results of mangrove area. Row (**a**) represents drone images; (**b**) represents the ground truth (GT); row (**c**–**h**) extraction results of different models. Columns (1–6) are diverse sites of drone images.

**Figure 12 sensors-25-02540-f012:**
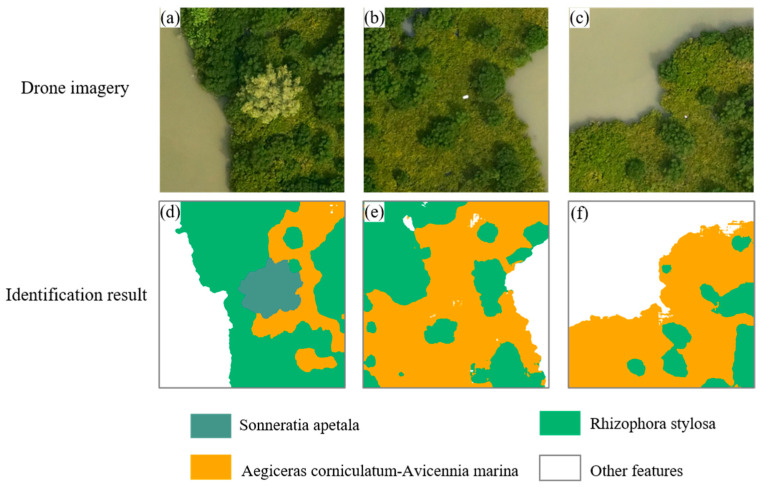
(**a**–**c**) Mangrove species images captured by drone; (**d**–**f**) identification results of mangrove species spatial distribution using AttCloudNet+.

**Figure 13 sensors-25-02540-f013:**
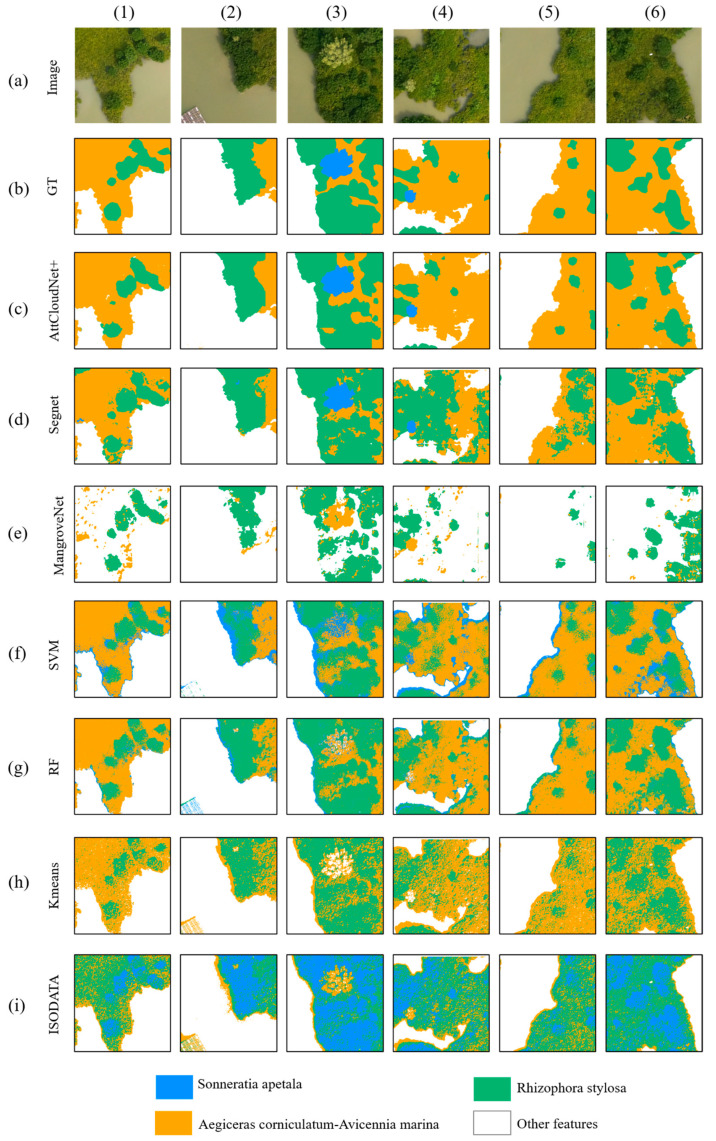
Comparison of different models’ prediction results of mangrove species. Row (**a**) represents drone images; row (**b**) represents the ground truth (GT); row (**c**–**i**) mangrove species identification results using different deep learning and conventional methods. Columns (1–6) are diverse sites of drone images.

**Figure 14 sensors-25-02540-f014:**
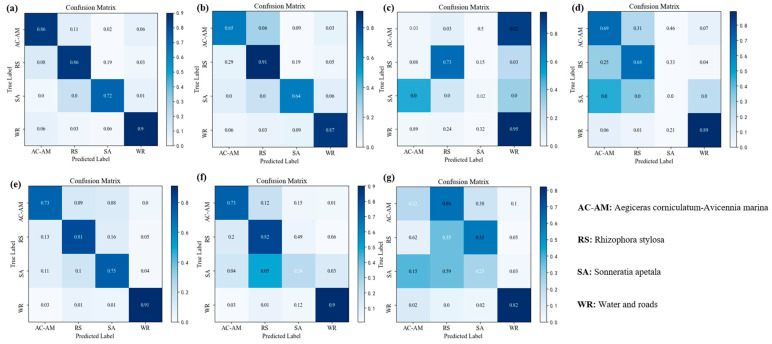
The confusion matrix of species identification results under different methods. The horizontal axis of each graph represents the category of the predicted results, and the vertical axis represents the category of the true value. (**a**) AttCloudNet+; (**b**) SegNet; (**c**) MangroveNet; (**d**) K-means; (**e**) SVM; (**f**) Random Forest; (**g**) ISODATA.

**Table 1 sensors-25-02540-t001:** Spectral range and resolution of the Phantom 4RTK multispectral drone sensor.

Spectral Name	Spectral Range and Tolerance (nm)	Spatial Resolution (In This Study) (cm)
Blue band (B)	450 ± 16	4.61
Green band (G)	560 ± 16	
Red band (R)	650 ± 16	
Red Edge band (RE)	730 ± 16	
Near Infrared (NIR)	840 ± 16	
Visible light band (RGB)	390~780	

**Table 2 sensors-25-02540-t002:** Kappa factor evaluation criteria [66].

Range of Coefficient Values	Evaluation Criteria
Kappa = −1	Completely inconsistent
Kappa = 0	Random classification results
0 < Kappa ≤ 0.2	Notably low consistency
0.2 < Kappa ≤ 0.4	Low consistency
0.4 < Kappa ≤ 0.6	Medium consistency
0.6 < Kappa ≤ 0.8	High consistency
0.8 < Kappa ≤ 1.0	Almost complete consistency
Kappa = +1	Complete consistency

**Table 3 sensors-25-02540-t003:** Performance of different deep learning models in mangrove area extraction.

Model Structure	Accuracy	F1_Score	mIoU	Precision	Recall
UNet	94.52%	96.20%	94.52%	94.52%	99.95%
DeepUNet	99.13%	92.10%	91.04%	93.15%	91.89%
ResUNet	97.13%	97.47%	96.00%	96.00%	99.95%
SegNet	96.71%	97.20%	95.63%	95.64%	99.95%
AttCloudNet+	95.18%	96.50%	94.97%	94.97%	99.96%
SegFormer	94.53%	96.20%	94.53%	94.53%	99.96%
MangroveNet	99.13%	98.84%	98.11%	99.62%	98.38%

**Table 4 sensors-25-02540-t004:** Precision comparison of mangrove species identification results by different methods.

Methods	Kappa Coefficient	Overall Accuracy (OA)
K-means	0.61	0.75
ISODATA	0.36	0.56
Random Forest	0.71	0.81
SVM	0.76	0.84
SegNet	0.73	0.82
MangroveNet	0.41	0.67
AttCloudNet+	0.81	0.87

**Table 5 sensors-25-02540-t005:** Results comparison of mangrove species identification model and artificial recognition.

Mangrove Species	AttCloudNet+ (m^2^)	Manual Delineation (m^2^)
*Aegiceras corniculatum*-*Avicennia marina*	6873.33	6013.33
*Rhizophora stylosa*	7960.00	7680.00
*Sonneratia apetala*	1386.67	1713.33

**Table 6 sensors-25-02540-t006:** Advantages and weaknesses of different models for mangrove area identification.

Model Structure	Advantage	Weakness
UNet	Lightweight model for simple segmentation tasks	The ability is slightly worse to process the complex high-resolution multi-spectral drone images
DeepUNet	Focus on the image’s spectral features	Ignored the spatial feature information
ResUNet	Lightweight model for simple segmentation tasks	Only considers spatial features and ignores spectral features
SegNet	The number of parameters is moderate, suitable for general use	The spatial feature information is ignored
AttCloudNet+	Mangrove area can be extracted effectively from multi-spectral drone images	Massive parameters require high computing power
SegFormer	The model is novel and transformer is used	It is better to process ordinary images than multispectral remote sensed images
MangroveNet	Mangrove area can be extracted effectively from multi-spectral drone images	The generalization needs to be improved

## Data Availability

The datasets presented in this article are not readily available because the data are part of an ongoing study.

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
