# Peer review of "An Approach for Detecting Mangrove Areas and Mapping Species Using Multispectral Drone Imagery and Deep Learning"

_sensors, 2025, doi:10.3390/s25082540_

Round 1
Reviewer 1 Report
Comments and Suggestions for Authors
Please see the attachment

Reviewer 2 Report
Comments and Suggestions for Authors
The paper is intersting since it face the specific problem of mangrove mapping by means of drone imagery. The over all content of the paper is almost clear, but the presented results and the methodology must be improved.
- page 4 line 148 the authors write; “This study acquired in-situ drone image data near Jinniu Island …” Please remove in-situ since data are acquired by a drone, so they are remote sensed
- page 4 lines 168-169: The authors write “with an all-in-one multispectral imaging system that integrates a visible camera and five multispectral cameras”. This statement is not correct. The instrument does no use 5 multispctral camera, but five monochromatic cameras each operating at a different wavelngths. Please change the statement taking into account the comment of the reviewer.
- page 5 Table 1: Details on the sensor are missing: Please insert details on the architecture of the sensor, on how the sensor is made. I suggest to insert a photo , in particular a photo that depicts how the various cameras are mounted
- page 9 lines 334 – 337 in the sentence (https://www.agisoft.com/, Agisoft Metashape allows the processing of images from visi- ble or multispectral cameras into dense point clouds, textured polygon models, georefer-enced true orthophotos and spatial information data in the form of Digital Surface Models, accessed on 28 May 2024) remove “Agisoft Metashape allows the processing of images from visible or multispectral cameras into dense point clouds, textured polygon models, georeferenced true orthophotos and spatial information data in the form of Digital Surface Models,)
- page 11 figure 5: Text in on box of the flowchart: replace "radiation correction" with "Radiometric correction"
- page 16 paragraph 4.1: For seeks of clarity, authors must list and cite all the models tested before discussing their performance
- page17 at line552 Loss (metric?) is cited but it is not described, I do not understand. Please clarify. Moreover ithe performances of Loss metric is not reported in table 3. Please modify the content of table 3 and correspondingly the text
- page 18 paragraph 4.2: For seeks of clarity, authors must list and cite all the models tested before discussing their performances
- page 20 Figure 12, Row b: Does GT stand for ground truth or is it a model? If it is a model please specify it.
- page 21 lines 635 - 639: The authors write: “Table 4 presents a comparative analysis of the AttCloudNet+ with traditional supervised classification, unsupervised classification, machine learning, and other methodologies, including the K-means clustering approach, ISODATA cluster analysis, and the Random Forest (RF) model. The performance of the Kappa coefficient and the overall accuracy (OA) index of the mangrove species identification by RF, Support Vector Machine (SVM), and other methods was evaluated.” It is not clear what the authors mean. Please rephrase the paragraph.
- page 22, Table 4: Table 4 reports comparison for only a set of the tested methods. Please explain why, or insert in table 4 all the tested methods.
- page 23, lines 706-707: The authors write “… and SegNet) for the mangrove area identification extraction, as shown in Table 3. These models….”. Please revise the format and remove the return .
- page 23 and 24, lines 708 – 715 : The authors write: “These models showed low efficiency, which may be affected by the following reasons: UNet had the lowest metric score. Due to the simple structure of the model, it may not be able to handle the complex texture features of high-resolution drone images. DeepUNet is a variant of UNet that considers the image's spectral features. However, the method ignored the spatial feature information, leading to uncertainty in identifying mangrove areas. SegNet has similar limitations to DeepUNet. Since ResUNet only considers spatial features and ignores spectral features, it may not perform as well as MangroveNet.” Please after column make a list of the different models with their pros ans cons
English language is poor and must be improved.
Going in to details:
- In general, throughout the paper, the autors use the term "association". This term is not appropriate. They have to change it, for example with the word "species",or remove it at all since it is not strictly necessary.
- page 3 line 100 English language: Use “remote sensed image” instead of “remote sensing image“
- page 3 lines 121-134: the description is confuse since the English language is poor . Moreover I sugget to use the present mode and no the past mode. i.e. “presents” and not “presented”; “provides” and not “provided”
- page 4 lines 153 and following: English language. The authors write: “These include the Avicennia marina group association, the Rhizophora stylosa group association, the Aegiceras corniculatum group association, the Sonneratia apetala group association, the Kandelia obovata group association, the Brugui era gymnorrhiza group association, the Acrostichum aureum group association, the Acan- thus ilicifolius group association, as well as the Excoecaria agallocha group association”. Not correct use of the term “association”. Please remove all "association" when describing Mangrove species.
- page 5 lines 186 and 187: lines 186 change the word “differential” with the word “different” and the word “association” with the word species
- page 5: English language : improve English in the caption of figure 2
- page 7 line 266: English language: in “here 𝑊𝑐𝑙 is the weights from …” it is not weights but wigth
- page 7 lines 269-271: English language Poor English please write again the sentence “However, given that the size of the feature layer is equivalent to that prior to the attention mechanism module, it is the feature layer after the maximum pooling" in order to make it clear and understandable
- page 10 and following pages, English language please change the word earth with Earth!!!!
- page 10 line 348: English language : it is not remote sensing image , but remote sensed image
- page 10 line 364: authors write: “Figure 5 shows the flowchart of this study. “. Flowchart is not described. Please describe in more detail the flowchart in Figure 5. Almost all the steps/boxes of the flowchart are not described in the paragraph.
- page 16 line 512, English language: In the sentence “ The explanation of TP, TN, FP, and FN shown in Figure 9 …. “add "are" before "shown"
- page 24 line 726: English language remove the words “This is because” at the beginning of the sentence.
Round 2
Reviewer 2 Report
Comments and Suggestions for Authors
The overall quality of the revised paper increased a lot, but some minor revisions are still needed, in particular for what concern English language.
Comments on the Quality of English LanguageThe following sentences have to be rewritten since their content is confuse and not understandable:
Pag. 2 47-49 : Given the extensive distribution of mangroves across the global coastal zone[4], monitoring the range and species distribution of mangroves with high spatial and temporal accuracy is positive for analyzing the changes in regional and even global marine ecosystems
Pag 5, caption of Figure 3: the authors write “Illustrates the features of different mangrove species in high-resolution drone images. (a) Rhizophora stylosa in drone image (left) and close-up photograph (right); (b) and (c) similar image cases of Aegiceras corniculatum and Sonneratia apetala.” Please remove “Illustrates the” at the beginning of the caption. The sentence must begin with “Features ….”
Pag. 12 lines 393 – 295 in the sentence “We chose this region because the mangrove forests here are lush and well-protected, making the boundary between the water and the mangrove areas more distinct” please remove the word “here”
